# Membrane Fouling Mitigation in MBR via the Feast–Famine Strategy to Enhance PHA Production by Activated Sludge

**DOI:** 10.3390/membranes12070703

**Published:** 2022-07-12

**Authors:** Santo Fabio Corsino, Gaetano Di Bella, Francesco Traina, Lucia Argiz Montes, Angeles Val del Rio, Anuska Mosquera Corral, Michele Torregrossa, Gaspare Viviani

**Affiliations:** 1Department of Engineering, Università di Palermo, 90128 Palermo, Italy; santofabio.corsino@unipa.it (S.F.C.); francesco.traina02@unipa.it (F.T.); michele.torregrossa@unipa.it (M.T.); gaspare.viviani@unipa.it (G.V.); 2Faculty of Engineering and Architecture, Università di Enna, 94100 Enna, Italy; 3CRETUS Institute, Department of Chemical Engineering, Universidade de Santiago de Compostela, 15782 Santiago de Compostela, Spain; luciaargiz.montes@usc.es (L.A.M.); mangeles.val@usc.es (A.V.d.R.); anuska.mosquera@usc.es (A.M.C.)

**Keywords:** extracellular polymeric substances (EPS), cake layer, fouling control, membrane bioreactor (MBR), polyhydroxyalkanoate (PHA), resistance in series (RIS) model, soluble microbial products (SMP)

## Abstract

Fouling is considered one of the main drawbacks of membrane bioreactor (MBR) technology. Among the main fouling agents, extracellular polymeric substances (EPS) are considered one of the most impactful since they cause the decrease of sludge filterability and decline of membrane flux in the long term. The present study investigated a biological strategy to reduce the membrane-fouling tendency in MBR systems. This consisted of seeding the reactor with activated sludge enriched in microorganisms with polyhydroxyalkanoate (PHA) storage ability and by imposing proper operating conditions to drive the carbon toward intracellular (PHA) rather than extracellular (EPS) accumulation. For that purpose, an MBR lab-scale plant was operated for 175 days, divided into four periods (1–4) according to different food to microorganisms’ ratios (F/M) (0.80 kg COD kg TSS^−1^ d^−1^ (Period 1), 0.13 kg COD kg TSS^−1^ d^−1^ (Period 2), 0.28 kg COD kg TSS^−1^ d^−1^ (Period 3), and 0.38 kg COD kg TSS^−1^ d^−1^ (Period 4)). The application of the feast/famine strategy favored the accumulation of intracellular polymers by bacteria. The increase of the PHA accumulation inside the cells corresponded to the decrease of EPS and an F/M of 0.40–0.50 kg COD kg TSS^−1^ d^−1^ was found as optimum to maximize the PHA production, while minimizing EPS. The lowest EPS content in the sludge (18% of total suspended solids) that corresponded to the maximum content of PHA (9.3%) was found in Period 4 and determined significant mitigation of the fouling rate, whose value was close to 0.10 × 10^11^ m^−1^ h^−1^. Thus, by imposing proper operating conditions, it was possible to drive the organic matter toward PHA accumulation. Moreover, a lower EPS content corresponded to a decrease in the irreversible fouling mechanism, which would imply a lower frequency of the extraordinary cleaning operations. This study highlighted the possibility of obtaining a double benefit by applying an MBR system in the frame of wastewater valorization: minimizing the fouling tendency of the membrane and recovery precursors of bioplastics from wastewater in line with the circular economy model.

## 1. Introduction

In the past years, membrane bioreactors (MBRs) have been widely applied in wastewater treatment due to the high quality of the effluent they generate and their low footprint [1]. In the framework of wastewater treatment intensification and the circular economy model, MBRs could play a crucial role. Indeed, because the membrane acts as a barrier for bacteria, MBRs could operate at higher biomass concentrations compared to conventional activated sludge systems, thus resulting in lower footprint requirements and the increase of the treatment capacities of existing wastewater treatment plants (WWTPs) at a given surface area [2]. Because of the positive role of membranes in the improvement of effluent quality, MBRs are often used for wastewater treatment or water reuse. Moreover, the complete bacterial retention noticeably improves the effluent quality, allowing the reclamation of the treated wastewater [3]. Additionally, MBRs reduce footprint significantly compared to the combination of tertiary treatments used for wastewater reuse, thus saving space for implantation and construction costs [4].

On the other hand, MBRs are increasingly applied in other biotechnological applications [5,6]. Among these applications, the production of polyhydroxyalkanoate (PHA), from renewable carbon sources, using open microbial cultures is one of the most explored biotechnological approaches since the possibility to recover added-value precursors to produce biomaterials alternative to fossil-based products [7]. This PHA production is generally carried out in a three-stage process that involves: substrate pre-treatment (i), enrichment of a culture in microorganisms with a high storage ability (ii), and accumulation to maximize intracellular compounds storage (iii) [8]. One of the main drawbacks of this process is the low volumetric productivity of PHA achieved by the low biomass concentration in operation in the enrichment and accumulation reactors. In this context, the application of a complete cell retention strategy was suggested as a feasible approach to tackle this issue [5]. Indeed, membrane retention would enable removal and addition of feed while keeping the cells inside the bioreactor, allowing obtaining a higher PHA daily production. Moreover, MBR would ensure high process performances especially if the enrichment phase is performed simultaneously with the wastewater treatment [9]. PHA accumulation in MBRs has been recently reported in the literature [5,10]. The authors successfully achieved the enrichment of a PHA producing culture in a submerged MBR system and achieved high PHA productivity (0.87–1.44 g PHA L^−1^ h^−1^). Similarly, in another study, high PHA productivity up to 4.6 g PHA L^−1^ h^−1^ was reported [6] using a pure culture and a selected substrate (i.e., glycerol, vegetable oils, etc.,). Despite the several advantages of MBRs, there are still drawbacks that should be addressed, among which the membrane fouling has been extensively studied due to its importance.

Membrane fouling in wastewater treatment systems consists of deposition of solid particles on the membrane surface or colloidal particles within the pores of the membrane [11]. This reduces the hydraulic performance of the membrane, causing a severe permeability decline or rapid trans-membrane pressure (TMP) increase. This determines the increase in maintenance and operating costs since the high energy consumption and frequent membrane cleaning operations [12,13]. Extracellular polymeric substances (EPS) were found to have a strong potential for membrane fouling. Several studies reported that the filterability of sludge decreased with the increase of EPS bound to the activated sludge flocs [14,15]. More precisely, these studies demonstrated that the specific cake resistance became higher as the specific amount of EPS increased [16,17]. Moreover, the hydrolysis of EPS could result in the increase of soluble microbial products (SMP) that have a remarkable effect on the irremovable membrane fouling mechanism [18]. Thus, limiting the EPS microbial synthesis could be a potential strategy to control the MBR biofouling. Overall, MBR fouling investigations still require a comprehensive understanding of the composition and fate of biopolymers that are responsible of fouling, as well as novel strategies aimed at minimizing the membrane fouling behavior.

EPS are excreted by bacteria and composed of a variety of organic substances, including proteins, carbohydrates, humic-like substances, etc., [19]. EPS are polymers biosynthesized by several strains of microorganisms in activated sludge systems, that triggered primarily a response to environmental stressors such as high salinity, presence of toxic substances, high/low temperature, high substrate availability, nutrient limiting conditions, etc., [20]. Moreover, EPS might act as carbon reserves to be used during substrate deficiency conditions [21]. In this sense, EPS production could be considered a parallel process to that of intracellular PHA accumulation when occurring under specific operating conditions such as high carbon availability and nutrients limitation, etc. Indeed, recent research studies carried out on mixed microbial culture (MMC) have demonstrated that EPS synthesis could lead to decreased PHA production since the competition for the available exogenous carbon source [9,22]. On the other hand, considering the impact of EPS on membrane fouling, reversing this competition toward PHA production could be a feasible approach to mitigate the membrane fouling. Although several biological-based strategies for membrane fouling control have been investigated in the past [23], the selective enrichment of the activated sludge with microorganisms having intracellular accumulation capacity, and the application of specific operating strategies aimed to drive the organic carbon into PHA rather EPS were not explored so far.

In this context, the present study aimed at evaluating the effects of promoting the PHA production by a MMC on the fouling behavior of an MBR ultrafiltration membrane. The main objective of this research work was to demonstrate that carbon conversion into PHA could hinder EPS synthesis and reduce membrane fouling. Moreover, membrane-fouling mechanisms were examined and correlated with the EPS content.

## 2. Materials and Methods

### 2.1. Set-Up and Operation of the MBR Plant

The MBR was inoculated with sludge collected from another enrichment system that had previously operated treating the same wastewater [9]. In this system, the activated sludge was successfully enriched with PHA producing microorganisms that were able to convert the organic carbon into intracellular storage compounds. For further details, the readers are referred to the literature [9]. The MBR consisted of a laboratory-scale reactor of 40 L working volume operating according to the aerobic dynamic feeding (ADF) regime [24]. This strategy consisted in the application of aerobic feast/famine cycles that provided for the alternation of excess and deficiency of carbon availability. The MBR was operated for 175 days divided into four periods (1–4) (Table 1).

Each period was characterized by different food to microorganisms’ ratios (F/M), which were equal to 0.80 kg COD kg TSS^−1^ d^−1^ (Period 1), 0.13 kg COD kg TSS^−1^ d^−1^ (Period 2), 0.28 kg COD kg TSS^−1^ d^−1^ (Period 3), and 0.38 kg COD kg TSS^−1^ d^−1^ (Period 4). The variation of the F/M ratio was obtained by increasing or decreasing the daily volume of wastewater treated or the biomass concentration in the reactor.

Specifically, in Period 1 (52 days) the MBR was fed with a flow of 30 L d^−1^ and was operated with a biomass concentration of 4 g TSS L^−1^. In Period 2 (63 days), the biomass concentration increased to 8 g TSS L^−1^ and the daily flow was reduced to 10 L d^−1^. In Period 3 (35 days) and Period 4 (25 days), the biomass concentration was maintained to approximately 8 g TSS L^−1^, while the daily flow was increased to 20 L d^−1^ and 30 L d^−1^, respectively. Operating conditions were changed once steady state in terms of PHA productivity was achieved and each period lasted at least three times the corresponding sludge retention time (SRT). The achievement of steady state was assumed when the ratio between the length of the feast phase and that of the entire cycle was below 0.20 according to the literature [25].

The MBR was a SBR (sequencing batch reactor) operated in cycles of 12 h consisting of 30 min of influent feeding, 600–640 min of aeration, 20–60 min of effluent discharge, and finally 30 min of idle. The volume of the raw wastewater fed at the beginning of the cycle was changed in each period in order to change the organic loading rate hence the F/M. The feeding time was maintained constant in each period, thus the volume of wastewater fed in each cycle was regulated by adjusting the flow rate of the feeding pump. Consequently, the hydraulic retention time and the volumetric exchange ratio were different in each period. The effluent discharge was performed by means of an ultrafiltration hollow-fibers membrane in submerged configuration (0.03 µm of porosity, 1.4 m^2^ of surface, PURON^®^ triple bundle Demo). The filtration cycle had a duration equal to 6 min, divided into 5 min of permeate extraction and 1 min of backwashing. The membrane backwashing was carried out by pumping a volume of permeate back through the membrane fibers from a clean in place (CIP) tank in which the permeate was stored. The membrane flux was maintained constant in all the experimental periods to approximately 14.57 L m^−2^ h^−1^ (suction flux) and the volume of permeate extracted during each period was regulated by adjusting the operating time of the suction pump during the cycle. The flux of permeate backwashing was equal to 8.57 L m^−2^ h^−1^ in all the periods. The reactor was equipped with a pair of porous stone diffusers placed at the bottom of the reactor that were connected to an air blower. Aeration was supplied during all stages excepting for the idle phase to allow complete oxygen depletion. All the equipment were connected to a programmable logic controller that handled the phases alternation. Figure 1 depicts a schematic representation of the reactor.

The MBR was fed with an industrial wastewater collected from a citrus processing industry. This effluent was characterized by high chemical oxygen demand (COD) concentrations close to 4500 g COD L^−1^. Table 2 summarizes the main characteristics of the wastewater.

Since the citrus wastewater lacked in nitrogen and phosphorous, a concentrated solution containing nitrogen and phosphorus was fed with the citrus wastewater at the beginning of the cycle to balance the ratio between total carbon (TC), as COD, total nitrogen (TN), and total phosphorus (TP). Urea and potassium-hydrogen phosphate were dosed to obtain a mass ratio between COD:TN:TP equal to 200:5:1. Moreover, pH was adjusted to neutral by adding sodium hydroxide.

The biomass concentration was maintained constant during each period by purging a known volume of sludge daily. Thus, the sludge retention time (SRT) was calculated by means of mass balances between the biomass in the reactor and that purged daily.

### 2.2. Analytical Methods

All the physical-chemical analyses of the liquid samples including total suspended solids (TSS), COD, TN, and TP were carried out according to standard methods [26]. Specifically, COD, TP, and TN were analyzed in the raw wastewater and permeate samples, without any pretreatment. Moreover, the COD was also measured in the supernatant of the mixed liquor at the end of the reaction cycle to assess the contribution of the biological process to COD removal without the membrane filtration.

The EPS were extracted from the activated sludge samples according to the literature [27]. Specifically, first the SMP and the loosely bound EPS were obtained by centrifuging an activated sludge sample at 5000 rpm for 5 min and filtering the supernatant with 0.22 µm membrane. Hereafter, the same sample was re-suspended with deionized water and heated in a water-bath at 80 °C for 10 min to allow solubilization of the flocs-bound EPS. Then, the sample was centrifuged at 7000× *g* rpm for 10 min at 4 °C and the supernatant filtered with 0.22 µm membrane to obtain the tightly bound EPS. Moreover, the EPS content was measured in the cake layer on the membrane fibers. This cake was manually removed from the fibers and collected in a glass beaker to obtain a homogeneous sample and it was characterized in terms of TSS and EPS concentrations. The EPS were extracted following the above-described method and then referred to the TSS concentration of the sample. The EPS were characterized by measuring the protein [28] and carbohydrates concentrations [29] using bovine serum albumin and glucose as standards, respectively. The activated sludge hydrophobicity was measured according to the literature [30], whereas the particle sized distribution (PSD) was assessed by means of an optic granulometer.

The PHA was extracted using 1–2 propylene-carbonate as a solvent [31]. Then, both the PHA and EPS content were expressed in dry weight referred to the TSS concentration (%TSS).

### 2.3. MBR Monitoring

Several operational cycles were measured throughout the operational period. Liquid and activated sludge samples were taken at precise time intervals during the cycle to measure the physical-chemical parameters and the contents of EPS and PHA.

To detect the end of the feast phase, the dissolved oxygen (DO) profile during the cycle was monitored. The end of the feast phase was identified by a rapid increase in the DO concentration in the reactor suggesting the complete depletion of the readily biodegradable organic substrate present in the liquid media. From the results of the analyzed liquid and biomass samples the different parameters used to characterize the reactor operation were estimated.

### 2.4. Membrane Fouling Analysis

The membrane fouling was investigated by assessing the total resistance to filtration (R_T_), the fouling rate (FR) and the specific deposition mechanisms according to the resistance in series (RIS) model [13]. Fouling characterization was performed at the end of each experimental period independently from the achievement of a maximum fouling rate or transmembrane pressure.

Briefly, the permeate flux and TMP were measured during normal plant operations, prior to cleaning, to assess the total resistance to filtration of the fouled membrane (*R*_*t*0_) according to the Equation (1):(1)$$R_{t0}=\frac{TMP_{1}}{J_{1}\cdot \mu }$$
where *J*_1_ [m^3^ m^−2^ s^−1^] and *TMP*_1_ [Pa] are the permeate flux and the transmembrane pressure measured at the end of the filtration cycle, respectively, while *μ* is the permeate viscosity [Pa s] at the operating temperature.

Before starting the MBR operation, the resistance to filtration due to the membrane (*R_m_*) was measured and this was assumed constant throughout the experiment. Hereafter, at the end of each period, the membrane was removed from the reactor and the cake manually detached by rinsing the fibers with tap water. Then, the membrane was immersed in clean water and the total resistance to filtration was measured (*R*_*t*1_). The membrane was chemically cleaned with a 5% sodium hypochlorite solution for 24 h at room temperature and then rinsed with tap water. The total resistance to filtration was measured again in clean water (*R*_*t*2_). Finally, the membrane was placed back in the bioreactor and subjected to the usual filtration cycle to evaluate the final total resistance to filtration (*R*_*t*3_). Thus, three different fouling mechanism were identified. First, the irremovable fouling (*R_irv_*), meaning the one that cannot be removed neither with physical nor chemical cleanings and that determines the decrease of the membrane permeability in the long-term operation. Second, the reversible fouling (*R_rev_*) meaning the fouling removable with ordinary backwashing. Third, the irreversible fouling (*R_irr_*) indicating the fouling removable only with extraordinary cleanings, physical or chemical.

The resistance to filtration due to these mechanisms are given by:(2)$$R_{irv}=R_{t2}-R_{m}\;\;\;\;\;\;$$
(3)$$R_{rev}=R_{t3}-R_{m}$$
(4)$$R_{irr}=R_{t0}-R_{t1}-R_{rev}+R_{irv}$$

For further details about the RIS model, the reader is referred to the literature [13].

## 3. Results and Discussions

### 3.1. Assessment of PHA Production in the MBR

The MBR was seeded with activated sludge previously enriched in PHA storing microorganisms. During the experiment, the MBR was operated under the feast/famine strategy to maintain the same conditions that enabled to promote the carbon conversion into PHA. The ratio between the duration of the feast phase and that of the entire cycle was monitored to assess the achievement of steady state that ensured the acclimation of the MMC to the specific operating conditions. When this ratio was below 0.20 it was considered that steady-state operating conditions were achieved, thus the MMC was acclimated to the specific operating conditions. At this time, the trends of the DO concentrations and PHA and total EPS contents in the bulk during the cycle were monitored in three consecutive days. Figure 2 depicts the DO concentration during a typical cycle, as well as that of PHA and total EPS measured in the bulk at steady state in each experimental period (average values). Figure 3 shows the DO concentration in relation to the soluble COD in the same operational days.

In Period 1 with a F/M ratio of 0.80 kg COD kg TSS^−1^ d^−1^, the operation of the reactor became stable after 37 days. On this day, the length of the feast phase was approximately 150 min, thus the ratio between the feast length and that of the whole cycle was 0.20. According to previous studies, this indicated that microorganisms developed a high intracellular storage capacity [32]. During the feast phase of Period 1, microorganisms converted the external substrate into both EPS and PHA. Indeed, the PHA content increased up to 7.4% of the TSS at the end of the feast phase, whereas EPS reached a maximum of 26.3%. At the end of the feast phase, not all the soluble COD was depleted and its concentration in the supernatant resulted close to 300 mg COD L^−1^, whereas at the end of the famine it was close to 100 mg COD L^−1^. During the famine phase both the EPS and PHA content decreased suggesting the occurrence of bacterial growth on external and internal storage compounds. However, bacterial growth occurred also on the residual soluble COD given that its concentration decreased during the famine phase.

In Period 2 with a F/M ratio 0.13 kg COD kg TSS^−1^ d^−1^, steady state operation was obtained after 58 days from the beginning of Period 2 (110th day of the experiment). The length of the feast phase decreased to 70 min because of the lower initial substrate concentration. Accordingly, the ratio between the length of the feast phase and the length of the entire cycle decreased to 0.10, thereby indicating the presence of microorganisms with storage capacity in the system. The maximum PHA content obtained at the end of the feast phase was 6.08%, whereas the EPS resulted close to 31%. Therefore, compared to Period 1 a slight increase in EPS content was observed at the expense of PHA.

In Period 3 (F/M ratio 0.28 kg COD kg TSS^−1^ d^−1^) after 28 days (143rd day) and Period 4 (F/M ratio of 0.38 kg COD kg TSS^−1^ d^−1^) after 23 days (173rd day), the maximum PHA content obtained at the end of the feast phase increased to 8.0% and 9.3%, respectively. In contrast, the EPS content showed an opposite trend since its content decreased to 23.8% in Period 3 and 19.8% in Period 4. In both these periods, the COD at the end of the feast phase increased compared to the previous periods. Specifically, the COD measured in the supernatant at the end of the feast phase increased to 110 mg L^−1^ and 190 mg L^−1^ in Period 3 and Period 4, respectively, whereas the values obtained at the end of the cycle were close to 50 mg L^−1^ and 100 mg L^−1^, thus indicating that the COD was not completely depleted during the feast phase (Figure 3).

Both the maximum PHA and the total EPS contents in the bulk showed a different relationship with the applied F/M in each experimental period (Figure 4). In more detail, it was observed that the synthesis of PHA increased to a maximum value in correspondence to a precise value of the F/M close to 0.40 kg COD kg TSS^−1^ d^−1^, after that it decreased as the F/M was increased. Conversely, the EPS decreased when the F/M increased reaching a minimum value, after that, it showed an increasing trend at F/M higher than 0.40 kg COD kg TSS^−1^ d^−1^.

The relationship observed between the PHA and the F/M suggested that high substrate availability was not favorable to promote the carbon storage into intracellular polymers. Indeed, when increasing the carbon availability at the beginning of the cycle, it was observed that it was not completely depleted during the feast phase (Figure 2a), thus bacterial growth could occur on external substrate also during the famine phase [33]. Thus, microorganisms with internal storage capacity were no longer favored over non-storing populations during famine phase, thus it is likely that they were gradually washed-out from the system [34]. In contrast, at lower F/M, the external substrate was completely depleted during the feast phase and converted into intracellular storage compounds (Figure 2b–d). Thus, microorganisms with internal storage capacity likely had a competitive advantage over non-storing populations during the famine phase since the latter had no available substrate for growth [35]. For this reason, the PHA production by the MMC decreased as the F/M increased from a certain value.

Moreover, the above results indicated that the increase of the PHA content corresponded to the decrease of that of EPS and vice versa. More precisely, the relationship observed between the PHA, EPS, and the F/M (Figure 4) suggested that very low and high substrate availability might be favorable to EPS-storing pathways, whereas at intermediate values of F/M (0.40–0.50 kg COD kg TSS^−1^ d^−1^) it was possible to promote the PHA storing, while minimizing that of EPS. The reasons for this result could be due to the loss of the competitive advantage of PHA-storing population over non-storing at high F/M as above discussed.

Another reason of this result could be that different bacterial strains are able to produce both EPS and PHA according to the specific operating conditions [36]. Therefore, at lower substrate availability it was likely that such microorganisms used shorter pathways to convert the organic substrate into external rather than internal storage compounds [37,38]. In this context, it is possible that the pathway for EPS production is less energy expensive, thus bacteria having both the capacity of storing EPS and PHA are more encouraged to store the organic carbon into extracellular compounds. Nevertheless, further studies are necessary to better clarify this.

Nonetheless, the above results indicated that it was possible to drive the organic matter conversion into PHA rather EPS by adjusting the F/M within a precise range of values. Therefore, it was possible to operate in such a way that the PHA content of the activated sludge was maximum while that of EPS was minimum. As it will be better elucidated in the following sections, this involved significant improvements in terms of membrane fouling.

### 3.2. COD Removal in the MBR

To evaluate separately the removal efficiency of the organic matter due to the biological process and the membrane retention, the COD concentration was measured in the supernatant of the mixed liquor at the end of the cycle and in the permeate. Figure 5 depicts the trends of the COD values obtained in the samples of supernatant and permeate at the end of the cycle throughout the four operational periods.

The influent COD concentration ranged between 4100 and 5150 mg L^−1^, which resulted in the typical range of citrus processing wastewater [39]. The trends of COD concentration in the supernatant and permeate at the end of the cycle were quite similar during the entire experiment. In more detail, in Period 1 the COD concentration in the supernatant and the permeate rapidly decreased to steady values close to 100 mg L^−1^ and 50 mg L^−1^, respectively, suggesting that steady-state was achieved at the end of the Period 1. In Period 2 and Period 3, the COD concentration further decreased to approximately 50 mg L^−1^ in the supernatant and 35 mg L^−1^ in the permeate. A slight increase was instead observed in Period 4, when the steady values of the COD in the supernatant was close to 100 mg L^−1^, whereas in the permeate it was approximately 50 mg L^−1^.

The results indicated that MBR was able to achieve very high COD removal, close to 99%. Specifically, the biological process accounted for about 97% of the COD removal, on average, whereas the membrane retention enabled to further increase the COD removal up to 99%. Thus, the results indicated a high biological activity of the biomass in the MBR that enabled an almost complete removal of the organic biodegradable fraction in the influent wastewater. Additionally, the membrane contributed to retain the supernatant turbidity as well as the residual particulate and inert COD that cannot be biologically degraded, thereby resulting in very high permeate quality eligible for reclamation as for the residual content of organic matter, according to the current European regulation [40]. Indeed, the effluent COD concentration was significantly lower for the limit of the discharge into the public sewer system (500 mg L^−1^) and it was lower even than that required for wastewater reclamation (100 mg L^−1^). Therefore, the MBR enabled to obtain very high effluent wastewater quality. Referring to other pollutants, such as nitrogen and phosphorus, their concentrations in the effluent were very low (<2 mg L^−1^) since these elements were supplied additionally to the wastewater in a quantity strictly necessary to the bacterial growth and proportionally to the COD removed.

The achieved results were in general in good agreement with previous applications of MBR for the treatment of high strength wastewater from food industry [39,41], thus confirming the high robustness of MBR toward the abatement of organic loading rates, both soluble and particulate, when treating high strength wastewater. In a previous study, a MBR treating citrus processing wastewater obtained approximately 98% of COD removal while operating at lower F/M (0.1–0.5 kg COD kg TSS^−1^ d^−1^) than the present study [39]. This suggested that operating with MMC enriched in PHA producing microorganisms also improved the removal of COD at high F/M, likely due to a higher metabolic activity of the biomass.

### 3.3. Characterization of EPS in the Bulk and Membrane Cake Layer

In the previous section, it was pointed out that the conversion of the organic carbon into PHA allowed for decreasing the total EPS content in the bulk since the storage of the external organic carbon was preferentially driven toward the intracellular accumulation. The EPS composition was assessed in the suspended bulk into the MBR and in the cake layer deposited on the membrane fibers (Figure 6).

The EPS fractions measured in the bulk of the MBR (Figure 6a) were mainly in the tightly bound form since the amount of SMP and loosely bound EPS were negligible during the entire experiment. Proteins were the main EPS constituent, accounting for approximately 80–85% of the total EPS, thus in agreement with other MBR studies, in which proteins were found to be the major constituents in EPS [42]. The total EPS content measured in the cake layer on the membrane fibers was comparable to that in the bulk in each period. The maximum and minimum contents were observed in Period 2 and Period 4, respectively, whereas intermediate values were measured in Period 1 and Period 3. However, it should be pointed out that the structure of the EPS in the cake layer significantly changed compared to that in the bulk. Indeed, the amount of loosely bound and SMP noticeably increased in the cake layer at the expense of the tightly bound fraction of the EPS. The ratio between the not-bound and the tightly bound EPS in the cake layer accounted for approximately 60–70%, thereby suggesting that a considerable amount of tightly bound EPS was hydrolyzed near the membrane fibers. The TSS concentration of the cake layer was significantly higher than the bulk (35–75 gTSS L^−1^). For this reason, it is likely that endogenous conditions occurred within the cake layer, since the very low F/M in this microenvironment. Therefore, the lack of external substrate led microorganisms to use the storage compounds to provide for carbon and energy supply to sustain the metabolic reactions. The release of the SMP and loosely bound EPS is recognized as one of the main membrane fouling agents. Several studies demonstrated that the increase of SMP in the bulk was associated with the occurrence of severe fouling in MBR [43]. Ding et al. (2020) reported that fouling tendency of the membrane started to worsen when the ratio between the loosely bound and bound-EPS increased [44]. Cosenza et al. (2013) suggested that SMP could pass through the membrane pores and they were gradually deposited and aggregated causing irremovable membrane fouling [15].

Nevertheless, it is worth noticing that in the present study the amount of SMP and loosely bound EPS decreased proportionally to that of the total EPS. Indeed, although the ratio between the soluble/loosely bound and the bound EPS was similar during each period, in terms of absolute value the amount of the not-tightly bound EPS was significantly lower in Period 3 and Period 4 when the total EPS content was minimum. Therefore, a lower EPS concentration corresponded to a lower concentration of not-tightly bound EPS hence the probability to encounter membrane fouling. This result, referring in particular to not-tightly bound EPS, suggests that minimizing the EPS content in the bulk could reduce the risk of SMP release that represents one of the most important causes of membrane fouling, especially referring to irremovable fouling deposition [17].

Table 3 summarizes the average values of the sludge hydrophobicity measured in the bulk and the results of the particle size distribution (PSD). The different EPS content in the activated sludge affected some of its physical properties, above all the hydrophobicity.

Indeed, the sludge hydrophobicity increased according to the average EPS content, resulting maximum in Period 1, when the EPS content in the sludge was the highest. Overall, the sludge hydrophobicity was slightly lower with that reported in previous studies on MBR and this was referred to the lower EPS content in the sludge [45]. Therefore, a lower EPS content corresponded to a lower sludge hydrophobicity. Referring to the PSD, no significant differences were noted between the four periods. Overall, the average size of the flocs (d_50_) was close to 50 µm, in line with the typical value.

### 3.4. Evaluation of Membrane Fouling Tendency and Mechanisms

The fouling magnitude and driving forces that led to the gradual loss in the membrane permeability depend on hydraulic parameters (i.e., critical flux) but also on the physical-chemical characteristics of the mixed liquor, such as the presence of foulant agents (i.e., EPS) [13]. Achieving sustainable operation means minimizing the magnitude of fouling, ensuring the maintenance of high permeate flux, and reducing the need for chemical cleaning operations. Figure 7 depicts the evolution of the membrane fouling expressed in terms of total resistance (Figure 7a), and fouling mechanisms according to the RIS model (Figure 7b).

The trend of total resistance was different in all the experimental periods (Figure 7a). Indeed, the fouling rate, represented by the increase of the total resistance per hour of filtration time, was close to 0.31 × 10^11^ m^−1^ h^−1^, 1.57 × 10^11^ m^−1^ h^−1^, 0.24 × 10^11^ m^−1^ h^−1^, and 0.11 × 10^11^ m^−1^ h^−1^ in Period 1, Period 2, Period 3, and Period 4 respectively. It is worth to pointed out that physical and chemical cleanings were performed at the end of each period to have a comprehensive characterization of the fouling mechanisms, although the total resistance to filtration was not such as to require extraordinary cleaning operations according to the membrane manufacturer’s suggestions (TMP < −0.50 bar). More precisely, the only period in which the TMP was close to the lowest value indicated in the membrane datasheet was Period 2, whereas, in the other periods, the highest TMP measured at the end of the filtration cycle was much lower. The lowest values of TMP were observed in Period 3 and Period 4 (−0.11 bar), thus confirming that the fouling tendency during these periods was very low. The application of the RIS model suggested that the impact of the fouling mechanisms was different in each period (Figure 7b). The main fouling mechanism was the irreversible cake deposition (*R_irr_*) in all the periods, whose contribution to the overall resistance to filtration ranged between 11 × 10^11^ m^−1^ (Period 4) and 52 × 10^11^ m^−1^ (Period 2). More precisely, the data above indicated that the higher the *R_irr_*, the greater the fouling rate and the loss in membrane permeability.

The resistance due to the reversible cake deposition (*R_rev_*) was significantly lower than the *R_irr_* and ranged between 4.9-10 10^11^ m^−1^, while showing a similar relationship with the fouling rate as observed referring to the *R_irr_*. The resistance due to irremovable cake deposition was very low during each period resulting close to 1.0 × 10^11^ m^−1^.

The above findings suggested a different tendency in the cake formation and deposition as the operating conditions in the MBR changed. Figure 8 reports the contribution of each specific fouling mechanism to the overall membrane-fouling tendency.

The contribution of the irremovable cake deposition was very low and slightly increased during the experiment reaching a maximum close to 6% in Period 4. Nevertheless, considering the low values of the total resistance due to the irremovable cake, this mechanism was not considered a key-foulant mechanism. The irremovable cake deposition was the most impacting mechanism. Indeed, it affected approximately 50–80% of the overall membrane fouling. It was interesting to note that its contribution significantly decreased during the experiment, reaching a minimum in Period 3 and Period 4 (50%). It was interesting to observe that the contribution of the reversible cake deposition increased against the trend of the irremovable cake. Indeed, the minimum contributions of the *R_rev_* (13–15%) were found in Period 1 and Period 2 when the contributions of the *R_irr_* were maximum, whereas the maximum contributions of the *R_rev_* (22–27%) were found in Period 4 and Period 3 when the contributions of the *R_irr_* were minimum (50%).

Comparing the results with that reported in previous studies on conventional MBR systems and others in which innovative solutions aimed at minimizing the fouling were evaluated, the magnitude of fouling was lower in the present work. Indeed, Di Trapani et al. [39] reported that the fouling rate of a hollow-fiber membrane, operating in the continuous mode under submerged configuration and treating the same wastewater of the present study ranged between 2.1–4.6 × 10^11^ m^−1^ h^−1^ (5–10 × 10^12^ m^−1^ d^−1^). In a recent study carried out on a MBR treating municipal wastewater, the authors observed a fouling rate between 0.40-3.30 10^11^ m^−1^ h^−1^ [46]. Hou et al. [47] obtained partial mitigation of the membrane fouling to approximately 0.42–0.53 × 10^11^ m^−1^ h^−1^ by the application of a micro-electric field. Furthermore, Souza et al. [48] reported that the application of electric field allowed the attenuation of the membrane fouling rate to 0.24 × 10^11^ m^−1^ h^−1^, which resulted comparable to the lowest values obtained in the present study although without using electric energy. Therefore, lowering the EPS content in the sludge by driving the conversion of the exogenous carbon toward PHA rather EPS resulted in a lower fouling tendency of the membrane.

### 3.5. Effect of PHA and EPS Content on Membrane Fouling

Based on the above results, it can be stated that the improvement of PHA production by the MMC limited the membrane-fouling tendency. Since the membrane operated at a constant flux of permeate during all the periods, it was speculated that the fouling was linked to the mixed liquor characteristics and to the EPS content. The total EPS and PHA content measured both in the bulk and in the cake-layer were correlated with the resistance due to irreversible and reversible cake depositions (Figure 9).

Generally, the correlation between the PHA and EPS content measured in the bulk and in the cake layer (EPS only) indicated that membrane fouling increased with the EPS content, whereas when the sludge was more enriched in PHA, the fouling tendency was mitigated. Considering both the EPS content in the bulk and the cake layer, they resulted well correlated with the reversible and irreversible cake deposition, thus confirming the role of the EPS in both the reversible and irreversible fouling mechanism (Figure 9a,b). Nevertheless, it is worth to be stressed that the ratio between the reversible cake deposition and the irreversible significantly decreased when the EPS content in the bulk and the cake was lower (Figure 9c). From an operating point of view, this implies that at lower EPS content the fouling removable with ordinary cleaning operations (i.e., permeate backwashing) increased. Therefore, the need for extraordinary cleaning operations (i.e., decommissioning, and physical cleaning of different membrane modules) aimed at removing the irreversible fouling is lower as the EPS content decreases. Furthermore, when the cake layer was more easily removable by ordinary backwashing, the concentration of TSS was lower. Indeed, from Period 2 to Period 4, the TSS concentration in the cake layer decreased from 73 g L^−1^ to 36 g L^−1^. This, in turn, reduced the risk of hydrolysis of the EPS hence the release of SMP that could increase the irremovable fouling, thereby hampering the long-term service life of the membrane.

The findings of this study suggested that operating with the activated sludge enriched in PHA storing microorganisms and by applying proper operating conditions that promote PHA production, involved a lower conversion of the organic carbon into EPS, which corresponded to a lower membrane-fouling tendency. Compared with other MBR operating under similar F/M, the EPS content in the mixed liquor in the present study was about 30–40% lower, on average [11,48,49]. Consequently, given the well-recognized role of the EPS in the membrane fouling, low content of such substances in the mixed liquor reduced the membrane-fouling tendency and did not lead to problems in membrane operation in the long term. Moreover, in contrast to what was observed in a previous study carried out on an MBR enriched in PHA accumulating organisms, cake formation was not an issue for hollow fibers membrane [5]. This difference was likely due to the sequencing operations implemented in the MBR in the present work. In general, the magnitude of fouling is lower in sequencing reactors, since the intermittent suction and membrane relaxation have been reported as the effective methods to reduce fouling and prolong membrane operation time and lifespan [49].

Based on the above results, it can be stated that a synergistic approach to the necessity of minimizing fouling tendency and the valorization of wastewater through its conversion into precursors of bioplastics could be obtained by coupling a PHA production process with the MBR technology. Further studies are necessary to optimize the process to maximize the enrichment of PHA-storing bacteria and the PHA recovery.

## 4. Conclusions

This study demonstrated feasibility in the stimulation of PHA production by a MMC as a possible strategy to mitigate membrane fouling in MBR systems.

By operating an MBR lab-scale plant within a certain range of F/M (0.40–0.50 kg COD kg TSS^−1^ d^−1^) exogenous carbon was preferentially converted into intracellular compounds maximizing PHA storage and minimizing EPS production.Lowering the EPS content of the sludge significantly decreased the fouling tendency of the membrane. The fouling rate considerably decreased reaching values below 0.2 × 10^11^ m^−1^ h^−1^. Moreover, this fact contributed to irreversible cake deposition.A lower EPS content corresponded to an increase of the fouling removable with ordinary backwashings, which could result in less frequent extraordinary cleanings operations.

## Figures and Tables

**Figure 1 membranes-12-00703-f001:**
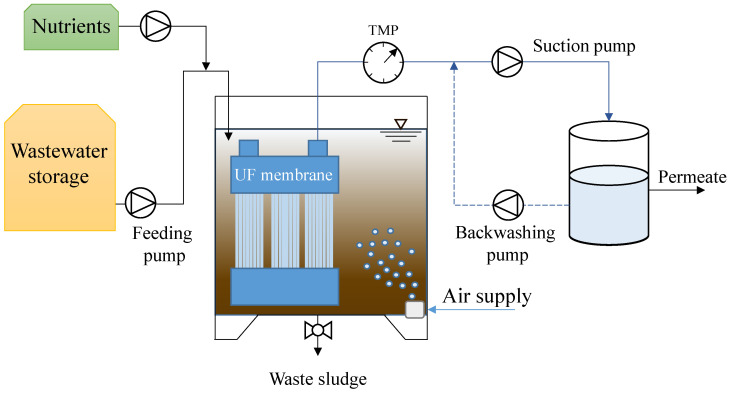
Layout of the membrane bioreactor (MBR).

**Figure 2 membranes-12-00703-f002:**
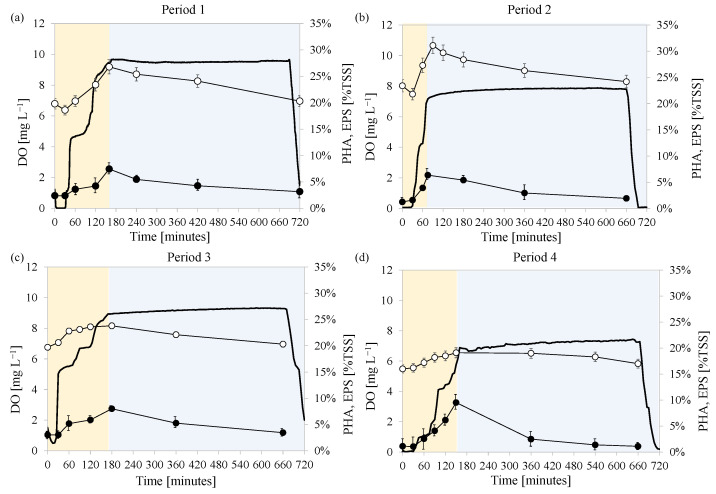
Trends of DO concentration (−), PHA (●) and total EPS content (○) of the activated sludge during feast (

) and famine (

) phases in each experimental period at steady state: after day 37 in Period 1 (**a**), after day 110 in Period 2 (**b**), after day 143 in Period 3 (**c**), and after day 173 in Period 4 (**d**).

**Figure 3 membranes-12-00703-f003:**
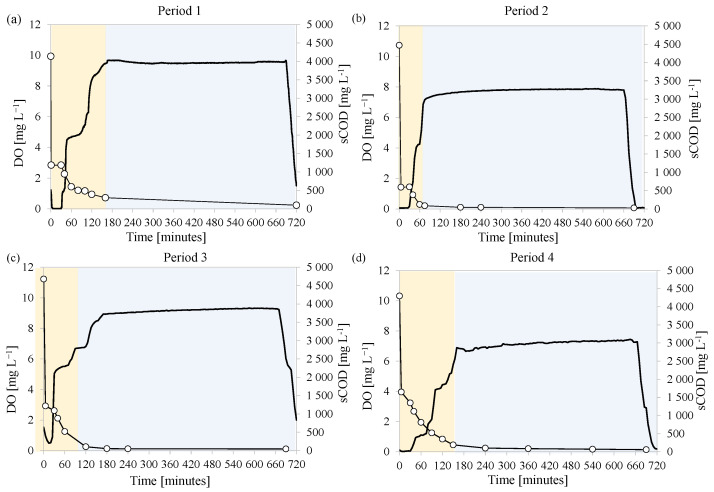
Trends of DO concentration (−) and soluble COD (○) during feast (

) and famine (

) phases in each experimental period at steady state: after day 37 in Period 1 (**a**), after day 110 in Period 2 (**b**), after day 143 in Period 3 (**c**), and after day 173 in Period 4 (**d**).

**Figure 4 membranes-12-00703-f004:**
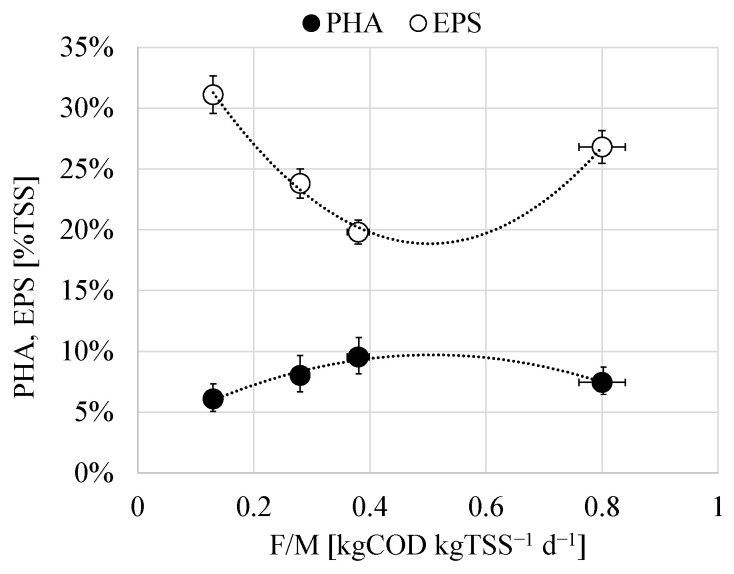
Relationship between the maximum total EPS in the bulk and PHA content with the F/M.

**Figure 5 membranes-12-00703-f005:**
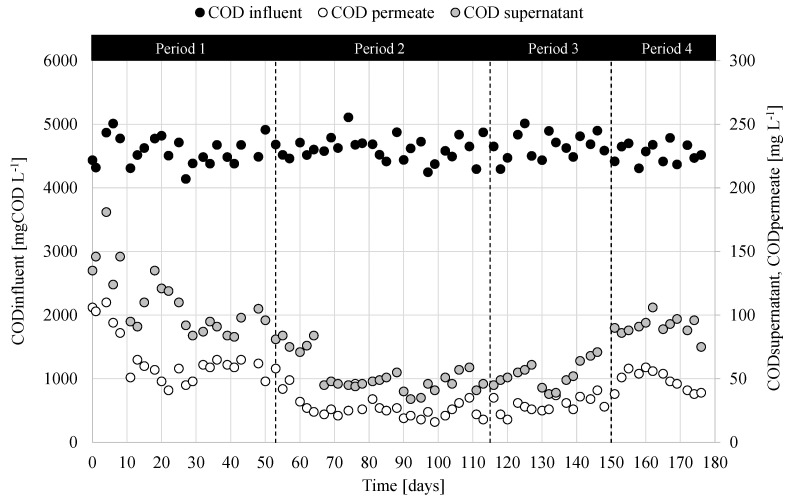
Trends of COD concentration in the influent wastewater, the supernatant, and permeate.

**Figure 6 membranes-12-00703-f006:**
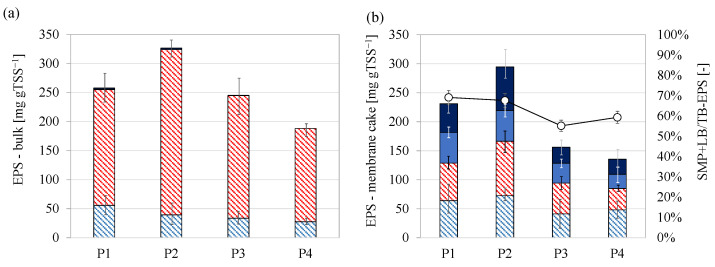
Values and composition of the total EPS measured in the bulk (**a**); value and composition of the EPS measured in the cake layer on the membrane and ratio between the not-tightly bound EPS (SMP + loosely bound) and tightly bound (TB)-EPS (**b**). Legend: tightly bound EPS as proteins (

), tightly bound EPS as carbohydrates (

), soluble and loosely bound EPS as proteins (

), soluble and loosely bound EPS as carbohydrates (

).

**Figure 7 membranes-12-00703-f007:**
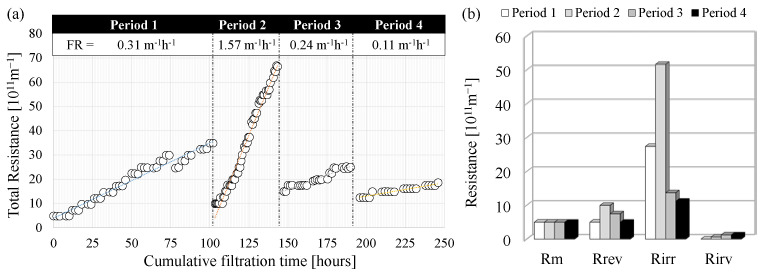
Trend of the total resistance to filtration and fouling rate during the experimental periods (**a**); results of the RIS model referring to the resistances to filtration (**b**).

**Figure 8 membranes-12-00703-f008:**
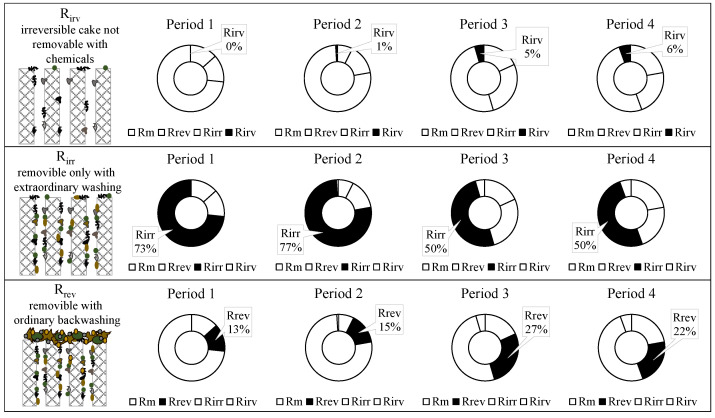
Contribution of the irremovable, irreversible, and reversible cake deposition on the membrane fouling.

**Figure 9 membranes-12-00703-f009:**
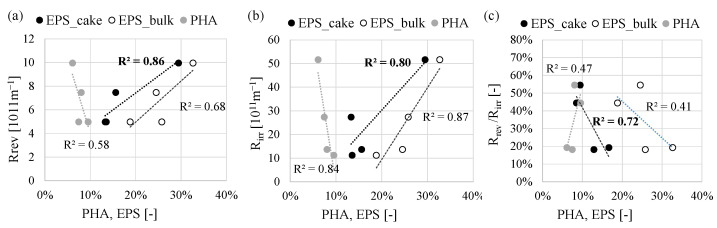
Correlations between the *R_rev_* (**a**), the *R_irr_* (**b**) and the ratio *R_rev_*/*R_irr_* (**c**) with the PHA and EPS content in the bulk (PHA and EPS) and the cake layer (EPS).

**Table 1 membranes-12-00703-t001:** Summary of the main operating conditions of the MBR.

	Period 1	Period 2	Period 3	Period 4
Operating days [d]	0–52	53–115	116–150	151–175
Daily flow [L d^−1^]	30	10	20	30
HRT [days]	1.33	4	2	1.33
Biomass concentration [g TSS L^−1^]	4.0 ± 0.2	8.0 ± 0.1	7.8 ± 0.3	8.2 ± 0.2
OLR [kg COD m^−3^ d^−1^]	3.2 ± 0.3	1.1 ± 0.2	2.2 ± 0.1	3.1 ± 0.2
F/M [kg COD kg TSS^−1^ d^−1^]	0.8 ± 0.2	0.13 ± 0.2	0.28 ± 0.2	0.38 ± 0.2
SRT [days]	7.4	24	10	8.9
Volumetric Exchange Ratio (VER) [%]	0.375	0.250	0.500	0.375
Hydraulic Retention Time (HRT) [d]	1.33	4	2	1.33
Temperature [°C]	17.8	18.3	19.4	21.0
Membrane suction flux [L m^−2^ h^−1^]	14.57	14.57	14.57	14.57
Membrane backwashing flux [L m^−2^ h^−1^]	8.57	8.57	8.57	8.57
Filtration time/cycle [min cycle^−1^]	60	20	40	60

**Table 2 membranes-12-00703-t002:** Characteristics of the raw wastewater used for the experiments.

Parameter	Unit	Value
Total COD	[mg L^−1^]	4558 ± 289
Soluble COD	[mg L^−1^]	3486 ± 201
Total nitrogen	[mg L^−1^]	12.6 ± 3.0
Total phosphorus	[mg L^−1^]	3.4 ± 0.7
pH	[-]	5.43 ± 0.49

**Table 3 membranes-12-00703-t003:** Average values of the activated sludge hydrophobicity (bulk) and characteristics diameters of the PSD.

Period	Hydrophobicity	PSD		
	[-]	d_10_ [µm]	d_50_ [µm]	d_90_ [µm]
Period 1	0.84 ± 0.04	21.3	44.12	81.41
Period 2	0.91 ± 0.02	26.4	51.68	96.44
Period 3	0.80 ± 0.06	19.5	60.47	95.13
Period 4	0.76 ± 0.05	16.3	55.59	90.71

## Data Availability

Data will be available on request to the corresponding author.

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
