# Peer review of "Membrane Fouling Mitigation in MBR via the Feast–Famine Strategy to Enhance PHA Production by Activated Sludge"

_membranes, 2022, doi:10.3390/membranes12070703_

Round 1

Reviewer 1 Report

This manuscript is aimed at evaluating the effect of enriching a microbial community with PHA accumulating organisms on the fouling behaviour of an MBR ultrafiltration membrane. The authors demonstrated that PHA synthesis could hinder EPS synthesis and mitigate membrane fouling. In this reviewer's opinion, while the findings are interesting and certainly has important implications for membrane-based bioreactors, this reviewer is not entirely convinced with the conclusions drawn by the authors based on the experimental approach.

More specifically, the authors found that PHA storage was higher at F/M ratios of 0.40 to 0.50 kg COD kg TSS-1 d-1, corresponding to lower EPS production. Based on this, the authors deduced that under such F/M conditions, the microbial community was enriched with PHA-storing microorganisms (lines 311 to 313). However, there was no genetic sequencing data showing that the microbial community had indeed shifted towards PHA-storing microorganisms, to support such a conclusion. Since some PHA-storing microorganisms are known to also produce EPS, it could have been equally likely that the same microorganisms are merely rewiring their metabolic pathways to redirect carbon flux away from EPS production towards PHA production. The authors would need to include microbial community analysis data, metagenomics and metatranscriptomics data to ascertain which of the above outcomes applied to the phenomenon they had observed. Without these data, the authors cannot conclude that PHA-producing or EPS-producing microbial population are enriched. As such, based on the data and conclusion provided by the authors, this reviewer cannot recommend this manuscript for publication at this stage.  

Author Response

Authors’ response: The Authors appreciated the Reviewer’s comment. The Authors agree with the Reviewer on the fact that specific microbiological analysis could provide more robust results to confirm the hypothesis of our study. As better explained in the revised manuscript, the MBR was inoculated with activated sludge that was previously enriched in PHA storing population in another SBR reactor. This was discussed in a previous study of the Authors that was added as reference for a better understanding (https://doi.org/10.1016/j.jwpe.2022.102772). In this study, we obtained the selective enrichment of the activated sludge with specific bacterial strains able to produce PHA. These data were strengthened by specific microbiologic analysis. Thus, in the present study we started up the MBR with a biomass already enriched in PHA storing microorganisms.

Nevertheless, the Authors would like to stress that the main focus of the paper was the analysis of the membrane fouling when the MBR was operated with a microbial community able to produce PHA. In our opinion, this is of greatest interest for the journal since it provides very useful information about a key topic of MBR. For this reason, the analysis of microbial community was out of the scope of the present study, but it will be widely discussed in a more specific manuscript for more specialist journals that will focus specifically on this aspect.

Our results demonstrated that the higher was the PHA produced by the microbial culture, the lower was that of the EPS and the fouling tendency. Thus, independently on the fact that the microorganisms are able to redirect carbon flux away from EPS production towards PHA production, we wanted to stress that if proper operating conditions are applied to select PHA-storing populations, or force the bacterial metabolism by stimulating intracellular storage, membrane fouling can be significantly mitigated

Author Response

Authors’ response: The Authors appreciated the Reviewer’s comment. In the revised version of the manuscript the Authors added some of the data requested by the Reviewer, such as the particles size distribution and the hydrophobicity of the mixed liquor and the cake layer. Unfortunately, the Authors are not able to provide data referring the Zeta potential of the mixed liquor. This aspect will be certainly considered in future studies on this topic. Referring the microbiological analysis, this was out of the scope of this study. Indeed, the main focus of the paper was the analysis of the membrane fouling when the MBR was operated with a microbial community able to produce PHA. In our opinion this is of greatest interest for the journal since it provides very useful information about a key topic of MBR. For this reason, the analysis of microbial community was out of the scope of the present study, but it will be widely discussed in a more specific manuscript for more specialistic journals that will focus specifically on this aspect.

1) It is not quite reasonable to use membrane separation in SBRs. Detailed explanations are needed to justify this design.

Authors’ response: In this study, the Authors wanted to evaluate the effect of activated sludge enrichment with microorganisms able to convert the organic matter into intracellular rather extracellular polymers on the fouling behaviour of an ultrafiltration membrane. Although the use of a membrane separation in SBRs is not usual, the reason of coupling the SBR and MBR technologies was that the enrichment of the activated sludge with PHA storing population is possible by applying a feeding strategy based on feast/famine phase’s alternation that is commonly achieved in SBR systems. Thus, it was necessary to couple the SBR technology with the MBR (and not the contrary) in order to obtain the enrichment of the activated sludge with PHA storing microorganisms.

2) Line 163-166. These sentences were repeated.

Authors’ response: The sentence was removed.

3) Section “3.1. Assessment of culture enrichment in the MBR”. The basis regarding the reactor becoming stable at 37 d, 110 d, 143 d, 173 d of the four periods, respectively, needs to be further justified. The concentration of EPS and PHA during the long-term operation of the MBR needs to be supplemented to better evaluate of culture enrichment process in the MBR. The variation of soluble COD concentration should be added in Fig. 2.

Authors’ response: The Authors considered that steady state conditions were obtained once the ratio between the duration of the feast phase and that of the entire cycle was below 0.20, as reported in most of the studies related to PHA accumulation using MMC. Since this aspect was widely reported in the literature, we used this control parameter to assess the achievement of steady conditions (very easy to be monitored) instead of performing the PHA and EPS measurement in the entire period to evaluate the enrichment process in the MBR.

A stable value of the feast duration was obtained at the 37 d, 110 d, 143 d, 173 d as reported in the manuscript. The Authors performed the analysis of the PHA and EPS during the cycle only at this phase of the experiment in three different days. The average data of EPS and PHA during the cycle are reported in Fig.2. The variation of the soluble COD concentration was reported in the new Figure 3.

4) Section “3.2. COD removal in the enrichment MBR”. As a wastewater treatment process, the effluent quality of an MBR should meet certain wastewater discharge standard. This has to be analyzed. Besides NH3-N, TN and TP, the most important water quality parameters, should be measured.

Authors’ response: The Authors improved the discussion adding more detail about the effluent discharge standards. Referring to other pollutants, such as nitrogen and phosphorus, the effluent concentrations were very low (< 2 mg L-1) since these elements were supplied additionally to the wastewater in a quantity strictly necessary to the bacterial growth and proportionally to the COD removed.

5) Section “3.3. Characterization of EPS in the bulk and membrane cake layer”. In Figure 5a, carbohydrates are the main component of the EPS, which is inconsistent with the description that proteins accounting for approximately 80-85% of the total EPS.

Authors’ response: The Authors apologize for the mistake. Indeed, the carbohydrates and proteins markers in the legend of figure 5 were wrongly reversed.

6) Proofreading and professional editing is needed.

Authors’ response: The paper was carefully revised according to the Reviewer’s suggestion

Reviewer 3 Report

In this manuscript, the authors investigated a biological strategy to reduce the membrane fouling tendency in MBR systems. This consisted of the enrichment of activated sludge in microorganisms with polyhydroxyalkanoate storage ability to drive the carbon toward intracellular rather than extracellular accumulation. The results were interesting and could be contributed to the MBR technology. As the authors declared that this study highlighted the possibility of obtaining a double benefit by applying an MBR system in the frame of wastewater valorization: minimizing the fouling tendency of the membrane and recovery precursors of bioplastics from wastewater in line with the circular economy model. Therefore, I suggest a minor revision of this manuscript. Detailed comments show below,

1.       “Keywords: EPS; cake layer; fouling control; MBR; PHA; RIS model; SMP”. Please avoid the abbreviations in Keywords!

2.       More illustrations on MBRs could be provided in the Introduction.

3.       Why did the DO concentration shown in Fig 2 almost vertically rise at 30 minutes and rapidly decrease at 660 min?

4.       “. Figure 4 depicts the trends of the COD values obtained in the samples of supernatant and permeate at the end of the cycle throughout the 4 operational periods.” It is not clear the CODin and CODout from Fig. 4. Could you provide more explanation?

Author Response

Authors’ response: The Authors thanks the Reviewer for the valuable comments. All the suggestions were considered and implemented in the revised version of the manuscript.

  1. “Keywords: EPS; cake layer; fouling control; MBR; PHA; RIS model; SMP”. Please avoid the abbreviations in Keywords!

Authors’ response: Edit as suggested.

  1. More illustrations on MBRs could be provided in the Introduction.

Authors’ response: The Authors added some more information about the MBR in general and the main factors affecting the fouling in the introduction.

  1. Why did the DO concentration shown in Fig 2 almost vertically rise at 30 minutes and rapidly decrease at 660 min?

Authors’ response: The aeration started after 30 minutes from the beginning of the cycle in order to start the biological reaction only once the feeding phase was completed. For this reason, no oxygen was supplied during this phase and, since no mixing devices were active, also the anaerobic reactions were negligible. To avoid having high oxygen concentration at the beginning of the cycle, the aeration supply was interrupted 30 minutes before the end of the cycle, to obtain complete oxygen depletion before the next cycle.

Reviewer 4 Report

A paper reports the application of alternating feast/famine stages in MBR to mitigate membrane fouling through the selection of conditions favorable for intracellular PHA storage microorganisms rather than EPS producing. The idea is obvious, since PHA storage is activated in microorganisms under starvation conditions, but it is difficult to implement it without reducing the organic load and, consequently, the efficiency of the bioreactor. Nevertheless, the authors were able to select the duration of the feasting and starvation stages in such a way as to achieve high overall COD removal and low membrane contamination. The manuscript is scientifically sound and the results are obtained using adequate methods and statistically treated. However, since there are no standard error bars in all the figures, it should be indicated how many experimental repetitions (similar runs of the bioreactor) were performed and how the results were statistically confirmed.

 Specific comments

1.      A title could be modified to show how the enrichment with PHA-storing microorganisms was achieved. Suggested: Membrane fouling mitigation in MBR via the feast–famine strategy to enrich PHA-storing microorganisms.

2.      It should be noted that many microorganisms can produce both PHA and EPS, which ratio can change depending on feeding conditions. Thus, it is rather the creation of conditions in the MBR for enhanced accumulation of PHA than the enrichment of microorganisms. This can be discussed further.

3.      Interestingly, both very low and high COD values stimulated EPS production and only quite narrow range of 0.40-0.50 kg COD kg TSS-1d-1 was favorable for the intracellular PHA storage. This finding, which is important for the MBR efficiency, can also be discussed further. It seems that the switch from feast to famine stage is acting as a stressor, which stimulates the PHA storage. Also, perhaps nitrogen and phosphorus availability plays an important role in this process.

Author Response

Authors’ response: The Authors thanks the Reviewers for the precious comments. Most of the analysis were performed in triplicates and in such cases, the error bars were added in the relative graphs.

 Specific comments

  1. A title could be modified to show how the enrichment with PHA-storing microorganisms was achieved. Suggested: Membrane fouling mitigation in MBR via the feast–famine strategy to enrich PHA-storing microorganisms.

Authors’ response: The Authors thanks the Reviewers for the suggestion. The title was modified according to the Reviewer’s comment.

  1. It should be noted that many microorganisms can produce both PHA and EPS, which ratio can change depending on feeding conditions. Thus, it is rather the creation of conditions in the MBR for enhanced accumulation of PHA than the enrichment of microorganisms. This can be discussed further.

Authors’ response: The Authors agree with the Reviewer’s comment. As better explained in the revised manuscript, the MBR was inoculated with activated sludge that was previously enriched in PHA storing population as was discussed in a previous study of the Authors (https://doi.org/10.1016/j.jwpe.2022.102772). In this study, we obtained the selective enrichment of the activated sludge with specific bacterial strains able to produce PHA. These data were strengthened by specific microbiologic analysis. Thus, in the present study we started up the MBR with a biomass already enriched in PHA storing microorganisms and evaluated how certain operating conditions affected the PHA (or EPS) production. Certainly, many of these bacteria can produce both PHA and EPS depending on the operating conditions, as demonstrated in the present study. However, the analysis of the MMC structure was out of the scope of this study, since the Authors wanted to demonstrate that the higher was the organic substrate converted into PHA, rather EPS, the lower was the membrane fouling tendency. The discussion was deepen in the revised version of the manuscript.

  1. Interestingly, both very low and high COD values stimulated EPS production and only quite narrow range of 0.40-0.50 kg COD kg TSS-1d-1 was favorable for the intracellular PHA storage. This finding, which is important for the MBR efficiency, can also be discussed further. It seems that the switch from feast to famine stage is acting as a stressor, which stimulates the PHA storage. Also, perhaps nitrogen and phosphorus availability plays an important role in this process.

Authors’ response: The Authors thanks the Reviewer for the comment. The feast/famine feeding strategy is a known stressor to stimulate PHA storage by bacteria, since at high substrate availability bacteria are not able to use all the available substrate for growth, thus the substrate is stored intracellularly during the feast phase and used during the famine. In the latter phase, since the lack of external substrate, only microorganisms that have stored the substrate are able to survive, thus the system will be gradually enriched with these microorganisms. Moreover, if nitrogen and phosphorous are not available, the growth process is hindered and the intracellular storage is more encouraged.

In this study, nitrogen and phosphorous were added in a precise amount in proportion to the COD (200:5:1) of the wastewater just to ensure the right quantity for the bacterial growth.

The conversion of the substrate into PHA or EPS depending on the operating conditions is a very complex topic. In the literature, there are several hypothesis about the mechanism involving the substrate utilization pathways that must be more in depth studied in more specific works. Certainly, specific microbiologic analysis could be helpful to this aim, but these were out of the scope of the present research.

Further details were added in the revised version of the manuscript.

Round 2

Reviewer 1 Report

The authors have improved the manuscript and in particularly, by providing more information about the type of seeding sludge (i.e., activated sludge previously enriched in PHA storing microorganisms), it was helpful for this reviewer to understand the characteristics of the microbial population in the reactor. Nevertheless, without microbial community analysis evidence, it would not be reasonable for the authors to conclude anything about the shift in microbial community from EPS-producing population to PHA-producing population and vice versa. Hence, the authors should refrain from such statements in the manuscript content and title, and instead, revise these statements to only talking about the changes/shift in EPS production and PHA production. The manuscript title should be revised accordingly as well. This would be more in line with the experiments conducted.

Author Response

The Authors appreciated the Reviewer’s comment. The title was modified to; “Membrane fouling mitigation in MBR via the feast–famine strategy to enhance PHA production by activated sludge” according to the Reviewer’s suggestion. Moreover, the manuscript was modified according to the Reviewer’s suggestions. Specifically, the discussion was changed by focusing about the shift in PHA and EPS production depending on the operating conditions.

Reviewer 2 Report

The revised version looks much better and can be recommended for publication in Membranes.

Author Response

The Authors appreciated the Reviewer’s comment. English language and style was carefully checked as requested by the Reviewer

Round 3

Reviewer 1 Report

The changes made to revise the conclusions, drawn from the experiments, were satisfactorily, and now are now in line with the experimental design. This reviewer has no further comments and recommends that this manuscript is acceptable for publication.